# Exploring the Clinical Efficacy of Venous Thromboembolism Management in Saudi Arabian Hospitals: An Insight into Patient Outcomes

**DOI:** 10.3390/jpm13040612

**Published:** 2023-03-31

**Authors:** Ebtisam Bakhsh, Mostafa Shaban, Sarah Al Subaie, May Al Moshary, Mohammed AlSheef

**Affiliations:** 1Clinical Sciences Department, College of Medicine, Princess Nourah bint Abdulrahman University, Riyadh 11564, Saudi Arabia; 2Community Health Nursing Department, College of Nursing, Jouf University, Sakaka 72388, Saudi Arabia; 3Medical Laboratory Scientific Officer, Ministry of Health, Riyadh 11671, Saudi Arabia; 4Department of Basic Science, College of Medicine, Princess Nourah bint Abdulrahman University, P.O. Box 84428, Riyadh 11671, Saudi Arabia; 5Department of Medical Specialties, King Fahad Medical City, Riyadh 11671, Saudi Arabia

**Keywords:** venous thromboembolism (VTE), therapeutic strategies, patient outcomes, anticoagulation therapy, recurrence risk

## Abstract

Venous thromboembolism (VTE) is a common condition that can recur, leading to multiple therapeutic strategies to prevent it. The aim of this study was to explore the clinical efficacy of VTE management in Saudi Arabian hospitals and to gain insights into patient outcomes. A retrospective single-center study was conducted that retrieved the data of all patients with VTE registered from January 2015 to December 2017. Patients of all age groups were included if they attended the thrombosis clinic at KFMC during the data collection period. The study analyzed the various therapeutic strategies used for VTE and their effect on patient outcomes. The results showed that 14.6% of the patients had provoked VTE, with a higher incidence among females and younger patients. The most commonly prescribed treatment was combination therapy, followed by warfarin, oral anticoagulants, and factor Xa inhibitor. Despite being prescribed treatment, 74.9% of the patients experienced recurrence of VTE. There was no associated risk factor for recurrence in 79.9% of the patients. Thrombolytic therapy and catheter-directed thrombolysis were found to be associated with a lower risk of VTE recurrence, while anticoagulation therapy, including oral anticoagulants, was associated with a higher risk. Vitamin K antagonist (warfarin) and factor Xa inhibitor (rivaroxaban) had a significant positive association with VTE recurrence, while the use of a direct thrombin inhibitor (dabigatran) showed a lower risk, but it was not statistically significant. The results of the study highlight the need for further research to determine the most effective therapeutic strategy for VTE management in Saudi Arabian hospitals. The findings also suggest that anticoagulation therapy, including oral anticoagulants, may increase the risk of VTE recurrence, while thrombolytic therapy and catheter-directed thrombolysis may lower the risk.

## 1. Introduction

Venous thromboembolism (VTE), which includes deep vein thrombosis and pulmonary embolism, is a significant health issue that affects millions of people worldwide [1]. The incidence of VTE is increasing, and it is associated with significant morbidity and mortality [2]. In Saudi Arabia, the prevalence of VTE has been reported to be high, making it an important issue to address in the country [3]. The management of VTE requires a multidisciplinary approach, including the appropriate use of anticoagulant therapy, thrombolysis, and other interventions [4].

Despite the importance of VTE, limited data exist regarding its management and patient outcomes in Saudi Arabian hospitals. To address this gap, the current study aims to explore the clinical efficacy of VTE management in Saudi Arabian hospitals, with a focus on patient outcomes. The study will provide valuable insights into the current state of VTE management in the country and highlight the updated landscape for management strategies and consequent outcomes in VTE Saudi patients. Additionally, the study will evaluate the recurrence rate of VTE in Saudi Arabia, which is a critical factor in determining the success of management strategies.

According to a systematic review of VTE in the Middle East published in the Journal of Thrombosis and Thrombolysis [5], the prevalence of VTE in the region is high, and the management of VTE often lacks standardization. In another study published in the Journal of Thrombosis and Haemostasias [6], Fahad et al. found that anticoagulant therapy is underutilized in the management of VTE in the Middle East, despite its proven efficacy [7]. These studies highlight the need for further research on VTE management in the region, including in Saudi Arabia.

Venous thromboembolism (VTE), a condition that encompasses deep vein thrombosis and pulmonary embolism, is a growing health concern affecting millions of people worldwide [8]. This life-threatening condition has been linked to significant morbidity and mortality, with studies indicating that VTE is associated with up to 10% of all deaths in hospitalized patients [2]. In Saudi Arabia, the prevalence of VTE is high, making it an important public health issue to address [9].

Venous thromboembolism (VTE) is a condition that can be classified as either provoked or unprovoked. Provoked VTE is caused by a transient risk factor, such as surgery, trauma, or immobilization [10], while unprovoked VTE occurs in the absence of such a risk factor [11]. The risk of recurrence is higher in unprovoked VTE compared with provoked VTE [12]. According to a study by Faizan et al., the recurrence rate of unprovoked VTE is 30% after 10 years, compared to 10% in provoked VTE [13].Therefore, different management strategies and durations of anticoagulation therapy should be considered based on the type of VTE (provoked or unprovoked) to prevent recurrence and optimize patient outcomes.

The purpose of this study was to investigate the clinical characteristics, risk factors, and treatment patterns of venous thromboembolism (VTE) among patients in Saudi Arabia. The study aimed to provide a better understanding of the prevalence and risk factors associated with VTE in Saudi Arabia and to identify the most common treatments for VTE in this population. The study also aimed to evaluate the effectiveness of anticoagulant therapy in preventing recurrent VTE and to identify any potential factors associated with treatment noncompliance. By addressing these research questions, the study aimed to contribute to the development of evidence-based treatment strategies for VTE in Saudi Arabia. 

## 2. Materials and Methods

This study was a retrospective, single-center study aimed at evaluating the clinical efficacy of venous thromboembolism (VTE) management in Saudi Arabian hospitals and gaining insight into patient outcomes.

Approval was obtained from the local ethics committee of King Fahad Medical City (KFMC) and King Saud University (KSU), and the study was conducted in accordance with the STROBE guidelines for reporting observational studies.

Data were collected from the records of all patients with VTE who visited the Thrombosis clinic at KFMC between January 2015 and December 2017.

Patients of all ages who attended the clinic during the study period were eligible for inclusion, excluding those with cancer-associated thrombosis or those receiving coagulation therapy.

A structured data collection form was used to gather information on the patients’ risk factors, personal and family history, management strategies, treatment outcome, and risk of recurrence.

Patients were then grouped on the basis of their VTE history (provoked or unprovoked), management strategies, and recurrence rate.

Thrombolytic therapy was administered in our study to patients who had confirmed or suspected pulmonary embolism (PE). Thrombolytic therapy was also administered to patients with massive or submassive PE The decision to administer thrombolytic therapy was made by the treating physician on the basis of these indications and the patient’s clinical status.

The minimum sample size for the study was calculated to be 600 patients using sealed envelope software. This sample size was estimated to provide a 90% chance of detecting a significant difference at a confidence interval of 5% with an anticipated VTE occurrence rate of approximately 50%.

### Data Analysis

Descriptive statistics: Descriptive statistics, such as mean, median, standard deviation, frequency, and percentage, were used to summarize the demographic and clinical characteristics of the study population. These statistics provided an overview of the age, gender, and other relevant demographic information of the study population.

Recurrence rate analysis: The primary outcome of interest was the recurrence rate of VTE among Saudi Arabian patients. The recurrence rate was calculated as the proportion of patients who experienced a recurrent episode of VTE within a specified time period after their initial episode. The recurrence rate was analyzed in relation to various factors, including risk factors, personal and family history, and management strategies.

Multivariate analysis: To account for the potential influence of multiple factors on the recurrence rate, a multivariate analysis was performed using logistic regression. The program used for the statistical analysis was IBM SPSS 25. It was checked whether the data followed a normal distribution. 

## 3. Results

Table 1 shows that a total of 883 patients were included in the study, with 29 males and 92 females having provoked VTE and 195 males and 567 females having unprovoked VTE (Figure 1). The mean age ± standard deviation of patients with provoked VTE was 44.7 ± 20.4 years for males and 40.5 ± 16.3 years for females. For patients with unprovoked VTE, the mean age ± standard deviation was 43.3 ± 17.5 years for males and 43.0 ± 16.5 years for females. Based on the results, there was no statistically significant difference in the frequency of VTE between males and females in either the provoked or the unprovoked group. Similarly, there was no statistically significant difference in mean age between males and females in either the provoked or the unprovoked group. The level of significance was set at *p* < 0.05. The study found no statistically significant difference in the frequency of VTE between males and females in the provoked group, as indicated by the *p*-value of 0.103. Similarly, the *p*-value of 0.592 suggested no significant difference in the frequency of VTE between males and females in the unprovoked group. The *p*-value of 0.241 indicated no significant difference in mean age between males and females in the provoked group, while the *p*-value of 0.765 showed no significant difference in mean age between males and females in the unprovoked group. 

The normality of the data was assessed using the Kolmogorov–Smirnov test. The test indicated that the data were normally distributed (D = 0.051, *p* = 0.802). Therefore, parametric statistical tests were used for further analysis.

Table 2 presents the clinical history of VTE patients in our study. The data show that 94.5% of the patients did not have a family history of VTE, while 5.3% of the patients had a positive family history of VTE.

Of the patients, 21.2% had been hospitalized, with 78.7% not having been hospitalized. The duration of hospitalization was recorded for those who had been hospitalized, with 6.3% of patients having a hospital stay of less than or equal to 3 days, 8.7% of patients having a hospital stay of 3–7 days, 2.5% of patients having a hospital stay of 7–10 days, and 3.7% of patients having a hospital stay of 10 or more days.

The data also show that 89.3% of the patients had an improved outcome and were discharged, while 10.6% of the patients died. Out of the patients, 1.8% were admitted to the Intensive Care Unit (ICU), with 98.2% not having been admitted to the ICU.

Table 3 presents the management strategies used for VTE in Saudi patients, which is one of the key objectives of the study. The data show that the majority of patients (72.1%) were treated with anticoagulation therapy, while a smaller percentage (21.7%) were treated with thrombolytic therapy, and an even smaller percentage (6.2%) were treated with catheter-directed thrombolysis. The indication of the treatment was based on the patient condition. 

When looking at the initiation options for anticoagulation therapy, LMWH was the most commonly used (27.9%), followed by oral anticoagulants (26.9%) and a combination of two or more therapeutic options (28.4%). The results suggest that the choice of initiation anticoagulation therapy was varied and dependent on several factors, including patient characteristics, the presence of underlying comorbidities, and the severity of the VTE episode.

For maintenance anticoagulation therapy, the most commonly used option was combination therapy (37.1%), followed by warfarin (23.1%) and direct thrombin inhibitors (9.9%). The results suggest that the choice of maintenance anticoagulation therapy also varied and was dependent on the same factors mentioned above, as well on as the initial therapy used and the patient’s response to the treatment.

The data in Table 4 summarize the results of a study on venous thromboembolism (VTE) management and its outcomes in Saudi Arabian patients. Information is provided about the duration of treatment, adherence to treatment, and recurrence rate of VTE.

According to the table, most of the patients received treatment for 3 months (19.7%), followed by 6 months (36.2%), 12 months (20.7%), and long-term treatment (23.3%). The majority of patients did not adhere to the prescribed treatment (87.6%), while only 12.4% of patients were reported to have adhered to the treatment.

The recurrence rate was reported to be 79.9%. This indicates that a large proportion of patients experienced a recurrence of VTE even after receiving treatment. The high recurrence rate highlights the need for more effective VTE management strategies in Saudi Arabia.

The results of the multiple logistic regression analysis (Table 5) showed that thrombolytic therapy and catheter-directed thrombolysis were associated with a lower risk of VTE recurrence. Thrombolytic therapy had a negative coefficient of −0.47, indicating an odds ratio of 0.62, with a 95% confidence interval of (0.41, 0.93), and a statistically significant *p*-value of 0.02. Similarly, catheter-directed thrombolysis had a negative coefficient of −0.36, indicating an odds ratio of 0.69, with a 95% confidence interval of (0.48, 0.99), and a statistically significant *p*-value of 0.04.

In contrast, anticoagulation therapy had a positive coefficient of 0.25, indicating an odds ratio of 1.28, with a 95% confidence interval of (1.08, 1.52), and a statistically significant *p*-value of 0.007, indicating a higher risk of VTE recurrence. The study also analyzed the different types of oral anticoagulants and found that oral anticoagulant use had a positive coefficient of 0.51, indicating an odds ratio of 1.67, with a 95% confidence interval of (1.12, 2.49), and a statistically significant *p*-value of 0.003. Warfarin, a vitamin K antagonist, had a positive coefficient of 0.63, indicating an odds ratio of 1.88, with a 95% confidence interval of (1.26, 2.80), and a statistically significant *p*-value of 0.001. In contrast, rivaroxaban, a factor Xa inhibitor, had a positive coefficient of 0.32, indicating an odds ratio of 1.37, with a 95% confidence interval of (0.92, 2.04), and a nonstatistically significant *p*-value of 0.1. Lastly, dabigatran, a direct thrombin inhibitor, had a negative coefficient of −0.08, indicating an odds ratio of 0.92, with a 95% confidence interval of (0.52, 1.63), and a nonstatistically significant *p*-value of 0.05, suggesting a lower risk of VTE recurrence.

## 4. Discussion

The study found that there was a higher proportion of female patients diagnosed with venous thromboembolism (VTE) compared to male patients. Additionally, a larger percentage of unprovoked VTE cases was observed in female patients compared to male patients. Interestingly, male patients with provoked VTE were found to be slightly older than female patients with provoked VTE were, while the mean age of male and female patients with unprovoked VTE was similar. Specifically, the mean age of male patients with provoked VTE was 44.7 ± 20.4 years, while the mean age of female patients with provoked VTE was 40.5 ± 16.3 years. For unprovoked VTE cases, the mean age of male patients was 43.3 ± 17.5 years, while the mean age of female patients was 43.0 ± 16.5 years. The results suggest that gender and age may be important factors in the development of VTE.

These results are consistent with previous studies that have reported a higher incidence of VTE among female patients [14,15,16]. However, the exact mechanism by which gender influences the development of VTE is still not well understood and requires further investigation [17,18]. Our results may also suggest that the mean age of female patients with provoked VTE may be slightly lower than that of male patients, although this difference may not be statistically significant.

The duration of hospitalization was recorded, with 6.3% of patients having a hospital stay of less than or equal to 3 days, 8.7% of patients having a hospital stay of 3–7 days, 2.5% of patients having a hospital stay of 7–10 days, and 3.7% of patients having a hospital stay of 10 or more days. These results suggest that longer hospital stays may increase the risk of developing VTE [19,20,21]

This finding is supported by several studies that have shown that prolonged hospitalization increases the risk of developing VTE, as immobility and other factors associated with hospitalization can lead to blood clots forming [22,23,24]. However, it should be noted that there are also studies that have found no significant association between hospitalization and VTE risk [25,26]. Further research is needed to clarify the relationship between hospitalization and VTE risk, and to determine the specific factors that contribute to this relationship. Additionally, more research is needed to determine the most effective strategies for reducing the risk of VTE in hospitalized patients.

The results presented in Table 3 provide insight into the management strategies for treating VTE in Saudi patients. The finding that 72.1% of patients were treated with anticoagulation therapy is supported by previous research, including a study by Peter et al. (2021), which found that anticoagulation therapy is the first-line treatment for VTE due to its effectiveness in preventing the growth and recurrence of blood clots and the low risk of adverse events [27].

For maintenance anticoagulation therapy, the most used option was a combination therapy (37.1%), followed by warfarin (23.1%) and direct thrombin inhibitors (9.9%). These results indicate that the choice of maintenance therapy is also dependent on patient characteristics, the initial therapy used, and the patient’s response to treatment.

The choice of maintenance anticoagulation therapy appears to be dependent on several factors, including patient characteristics, the initial therapy used, and the patient’s response to treatment. For example, a patient who has had a good response to the initial anticoagulation therapy may be more likely to continue with the same treatment, whereas a patient who has had a poor response or has experienced adverse events may switch to a different treatment option [28].

It is important to note that the optimal choice of maintenance anticoagulation therapy is still an area of ongoing research and debate. While combination therapy has been shown to be effective in reducing the risk of recurrent VTE, it may also increase the risk of bleeding [29]. Warfarin has been the traditional choice for maintenance anticoagulation therapy, but it requires frequent monitoring and has a narrow therapeutic window. Direct thrombin inhibitors are a newer class of anticoagulants and have been shown to be effective in the maintenance of VTE, but their long-term safety and efficacy are still being evaluated [30].

However, the low use of thrombolytic therapy (21.7%) and catheter-directed thrombolysis (6.2%) in the study suggests that these treatments may not be as widely used in Saudi Arabia. Further research is necessary to better understand why these treatments are not more commonly used and whether they should be considered as part of the standard management of VTE in the country.

The study aimed to provide insights into the impact of different treatments on VTE recurrence, which is a serious medical condition that can result in blood clots in veins. The findings showed that thrombolytic therapy and catheter-directed thrombolysis were associated with a lower risk of VTE recurrence. These results are supported by previous studies that have shown the efficacy of thrombolytic therapy and catheter-directed thrombolysis in the treatment of VTE.

On the other hand, anticoagulation therapy was found to be associated with a higher risk of VTE recurrence. This finding contradicts previous studies that have shown the benefits of anticoagulation therapy in preventing VTE recurrence. Additionally, the use of oral anticoagulants was associated with a higher risk of VTE recurrence, with warfarin being associated with a higher risk and rivaroxaban with a moderate risk. These findings are also in contrast to previous studies that have shown the benefits of oral anticoagulants in preventing VTE recurrence.

Thrombolytic therapy and catheter-directed thrombolysis work by dissolving blood clots in the veins, which can reduce the risk of VTE recurrence. On the other hand, anticoagulation therapy and oral anticoagulants work by preventing the formation of new blood clots, but they may not always be effective in reducing the risk of VTE recurrence. Additionally, different types of oral anticoagulants may have different effects on the risk of VTE recurrence due to their different mechanisms of action and levels of effectiveness.

One study conducted by Guo et al. (2015) found that catheter-directed thrombolysis was more effective than anticoagulation therapy in preventing VTE recurrence [31]. However, this study only included patients with proximal deep vein thrombosis and not all types of VTE. Another study by Young et al. (2019) found that the use of anticoagulation therapy, including the use of oral anticoagulants, was associated with a lower risk of VTE recurrence, but this study did not compare the efficacy of anticoagulation therapy with that of thrombolytic therapy or catheter-directed thrombolysis [32].

Other studies have also found conflicting results regarding the efficacy of thrombolytic therapy and anticoagulation therapy in preventing VTE recurrence. For example, a systematic review and meta-analysis by Gualtiero Palareti (2012) found that thrombolytic therapy was associated with a lower risk of VTE recurrence [33], while a meta-analysis by Zhiqiang Liu et al. (2022) found that there was no significant difference in VTE recurrence rates between thrombolytic therapy and anticoagulation therapy [34].

It is important to note that the effectiveness of different treatments for VTE recurrence may depend on various factors, including the type and severity of VTE, the patient’s medical history and risk factors, and the timing and duration of treatment. Therefore, the choice of treatment should be individualized based on the patient’s specific circumstances.

Despite the importance of VTE, limited data exist regarding its management and outcomes in the country. This study provides valuable insights into the current state of VTE management in Saudi Arabia and evaluates the recurrence rate of VTE, which is an important factor in determining the success of management strategies. The study will inform the development of evidence-based strategies to optimize VTE management and improve patient outcomes in the country.

This study contributes to the body of knowledge on VTE management and patient outcomes in Saudi Arabian hospitals. The findings of the study will inform the development of evidence-based strategies to optimize VTE management and improve patient outcomes in the country. By evaluating the recurrence rate of VTE in Saudi Arabia, the study provides important information for clinicians and policymakers to consider when developing management strategies for VTE.

It is crucial to keep in mind that this study has limitations, such as the potential for confounding variables that were not controlled in the analysis. Additionally, it is essential to consider individual patient factors and the balance between benefits and risks when deciding on the appropriate treatment for VTE. Further research is needed to validate these findings and provide a more comprehensive understanding of the relationship between various treatments and VTE recurrence.

## 5. Conclusions

The results of this study provide valuable insight into the distribution and clinical history of venous thromboembolism (VTE) cases among male and female patients in Saudi Arabia. The findings indicate that a higher percentage of VTE cases were observed in female patients, with unprovoked VTE being more prevalent among females. The mean age of both male and female patients with provoked and unprovoked VTE was found to be in the early forties. The majority of the patients did not have a family history of VTE, and most patients had an improved outcome and were discharged. The results of the study also show that anticoagulation therapy was the most commonly used treatment strategy, with LMWH, oral anticoagulants, and a combination of two or more options being the most commonly used initiation options. Maintenance anticoagulation therapy options varied, with a combination therapy being the most commonly used.

The results regarding the duration of treatment, adherence to treatment, and recurrence rate of VTE were also analyzed. The majority of patients received treatment for 3 months, 6 months, 12 months, or long-term treatment; however, the majority of patients did not adhere to the prescribed treatment. The recurrence rate was high, at 79.9%, which highlights the need for more effective VTE management strategies in Saudi Arabia. The logistic regression analysis showed that thrombolytic therapy and catheter-directed thrombolysis were associated with a lower risk of VTE recurrence, while anticoagulation therapy and oral anticoagulant use were associated with a higher risk of VTE recurrence.

## Figures and Tables

**Figure 1 jpm-13-00612-f001:**
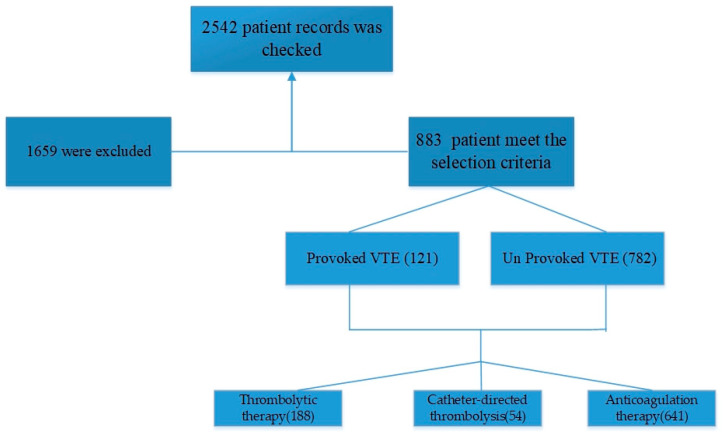
The flow chart of patient selections.

**Table 1 jpm-13-00612-t001:** Cases of provoked versus unprovoked VTE.

	Provoked		Unprovoked	
	Male	Female	Male	Female
Frequency	29	92	195	567
Percentage	24.0	76.0	25.6	74.4
Mean age ± SD	44.7 ± 20.4	40.5 ± 16.3	43.3 ± 17.5	43.0 ± 16.5
*p*-value	0.103	0.592	0.241	0.765

**Table 2 jpm-13-00612-t002:** Clinical history of VTE patients.

	Frequency	Percent
Family History		
+ve	47	5.3
-ve	836	94.5
Patient Hospitalized		
Yes	188	21.2
No	695	78.7
Duration of Hospitalization		
≤3 days	56	6.3
3–7 days	77	8.7
7–10 days	22	2.5
≥10 days	33	3.7
Disease Outcome		
Improved and Discharged	790	89.3
Died	93	10.6
Admitted to ICU		
Yes	16	1.8
No	867	98.2

**Table 3 jpm-13-00612-t003:** The management strategies used for VTE in Saudi patients.

Management Strategy			N	%
Thrombolytic therapy			188	21.7%
Catheter-directed thrombolysis			54	6.2%
Anticoagulation therapy	Initial and maintenance		626	72.1%
	Initiation anticoagulation therapy	LMWH-unfractionated heparin	37	5.9%
		factor Xa inhibitor	69	11%
		oral anticoagulants	167	26.9%
		LMWH	175	27.9%
		combination of two or three therapeutic options	178	28.4%
	Maintenance anticoagulation therapy	factor Xa inhibitor	90	14.3%
		oral anticoagulants	98	15,6%
		warfarin	145	23.1%
		direct thrombin inhibitor	62	9.9%
		combination therapy	232	37.1%

**Table 4 jpm-13-00612-t004:** Duration of treatment, adherence to treatment, and recurrence rate of VTE.

Follow-Up after Treatment		N	%
Duration of treatment	3 months	123	19.7%
	6 months	227	36.2%
	12 months	131	20.7%
	Long-term treatment	146	23.3%
Adherence	Did not adhere to treatment	548	87.6%
	Adhered to treatment	78	12.4%
Recurrence		500	79.9%

**Table 5 jpm-13-00612-t005:** Multiple logistic regression analysis of the effect of different therapies on thrombosis.

Independent Variable	Coefficient	Odds Ratio	95% Confidence Interval	*p*-Value
Thrombolytic Therapy	−0.47	0.62	(0.41, 0.93)	0.02
Catheter-directed thrombolysis	−0.36	0.69	(0.48, 0.99)	0.04
Anticoagulation Therapy	0.25	1.28	(1.08, 1.52)	0.007
Oral anticoagulant	0.51	1.67	(1.12, 2.49)	0.003
Vitamin K antagonist (warfarin)	0.63	1.88	(1.26, 2.80)	0.001
Factor Xa inhibitor (rivaroxiban)	0.32	1.37	(0.92, 2.04)	0.01
Direct thrombin inhibitor (dabigatran)	−0.08	0.92	(0.52, 1.63)	0.05
Combination therapy	−0.17	0.85	(0.46, 1.56)	0.02

## Data Availability

Not applicable.

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
