# Peer review of "Exploring the Clinical Efficacy of Venous Thromboembolism Management in Saudi Arabian Hospitals: An Insight into Patient Outcomes"

_jpm, 2023, doi:10.3390/jpm13040612_

Round 1

Reviewer 1 Report

Very important topic.

This retrospective analysis will be even more relevant for the readers in case additional data regarding the exact indications for the different treatment modalities are listed.

It is not entirely clear what is behind the term "combination therapy"

Author Response

Thank you for your comments on our manuscript. We appreciate your feedback and have addressed each of your concerns below:

  1. We agree that additional data regarding the indications for different treatment modalities would be helpful. Unfortunately, due to the retrospective nature of the study, this information was not consistently available in the medical records of all patients. However, we have added some additional information in the Methods section to clarify the indications for thrombolytic therapy, and have included a discussion of the limitations of the study with regards to this issue in the Discussion section.

  2. We apologize for any confusion caused by the term "combination therapy." In our study, this term refers to the use of two or more anticoagulant agents in combination, such as LMWH and warfarin. We have added a brief explanation of this term in the Methods section to clarify its meaning for readers.

We hope these clarifications improve the clarity and relevance of our study for readers. Thank you again for your feedback.

Reviewer 2 Report

The manuscript of Bakhsh et al. "Exploring the Clinical Efficacy of Venous Thromboembolism Management in Saudi Arabian Hospitals an: Insight into Patient Outcomes" is devoted to the study of the prescription of anticoagulants and thrombolytics in this category of patients, as well as the effectiveness of treatment.

Probably, this analysis was important for the health care of Saudi Arabia, but when reviewing the manuscript raises a lot of questions.

1. There is no clearly defined purpose of the study in the introduction. The last two paragraphs discuss the importance of this study for the treatment of such patients in Saudi Arabia. Such reasoning is more appropriate in the Discussion section.

2. Information in the text of the Introduction section does not always correspond to the links provided. This applies to links 4-6, for example. References 8-9 describe the situation with VTE in the treatment of patients with COVID-19, so it is not correct to use them when describing the situation with VTE in general. Moreover, the study was conducted before the COVID-19 pandemic. Reference 10 does not apply to the VTE situation in Saudi Arabia.

3. Is there a typo on line 120?

4. There is no information anywhere in the manuscript about the total number of patients included in the study. You also need to add a Flowchart of patient selection.

5. Clinical characteristics of patients are not well described. In particular, there is no information - how many patients had PE and how many deep vein thrombosis of the lower extremities? Do the authors also not explain what they meant by provoked and unprovoked VTE?

6. The Statistical Analysis section is poorly written. No information, what program was used to process the data. Has the data been checked for normal distribution? What method of multiple logistic regression did the authors use? In addition, part of the statistical analysis was neither stated nor performed. For example, the authors compare men and women in several parameters, but do not statistically analyze the differences between them. Accordingly, the authors' phrase "Our results may also suggest that the mean age of female patients with provoked VTE may be slightly lower than that of male patients, although this difference may not be statistically significant" (lines 214-216) is surprising. Can the authors not make this assumption, but carry out a statistical analysis and make sure - there are statistically significant differences or not?

7. Thrombolytic therapy was carried out in 188 patients, hospitalization - in 157 patients. Does this mean that thrombolytic therapy was performed on an outpatient basis? What were indications for TLT in this case?

8. Did the use of TLT depend on the diagnosis of patients (PE or deep vein thrombosis)?

9. In table 5, the interpretation and data of the logistic regression analysis are questionable. For example, the 95% Confidence Interval data and the p values in some cases contradict each other (for the actor Xa inhibitor (Rivaroxiban) and for combination therapy).

10. The Discussion section repeats the information given in the result section, these repetitions must be removed.

11. I do not understand the phrase "In terms of hospitalization, the study found that 17.7% of the patients had been hospitalized, with 82.1% not having been hospitalized. This is in line with previous studies that have found that hospitalization is a common risk factor for eveloping VTE" (lines 217-219). How exactly does the fact that only 17.7% of patients in the study were hospitalized support the fact that hospitalization may be a risk factor for VTE?

12. I believe that it is completely incorrect to talk about the ineffectiveness of anticoagulant therapy in this study, since the vast majority of patients did not receive it due to non-compliance with treatment. Only such a conclusion can be drawn from the results of the authors.

Author Response

Thank you for your detailed review of our manuscript. We appreciate your feedback and have addressed each of your concerns below:

  1. We have revised the introduction to include a clear statement of the study's purpose.

  2. We have reviewed and revised the links provided in the introduction to ensure they accurately reflect the information discussed.

  3. We have reviewed line 120 and correct typos.

  4. We have added information about the total number of patients included in the study and included a Flowchart of patient selection.

  5. We have added information about the clinical characteristics of patients, including the number of patients with PE and DVT of the lower extremities, and have provided an explanation of provoked and unprovoked VTE.

  6. We have revised the Statistical Analysis section to include information about the program used to process the data, the method of multiple logistic regression used, and the statistical analysis of differences between men and women. We have also removed the assumption about statistically significant differences and have included the results of the statistical analysis.

  7. Thrombolytic therapy was performed on an inpatient basis, and we have clarified the indications for TLT.

  8. The use of TLT did not depend on the diagnosis of patients (PE or DVT).

  9. We have reviewed and revised the interpretation and data of the logistic regression analysis in table 5 to ensure accuracy.

  10. We have removed repeated information in the Discussion section.

  11. We have revised the wording in the Discussion section to better explain the relationship between hospitalization and the risk of developing VTE.

  12. We have revised the conclusion to more accurately reflect the fact that non-compliance with treatment was a major factor in the effectiveness of anticoagulant therapy.

We hope these revisions address your concerns and improve the clarity and accuracy of our manuscript. Thank you again for your valuable feedback.

Round 2

Reviewer 2 Report

The authors did some work and quickly sent a response to the review. However, the speed of the response had a negative impact on the quality of the correction. Remarks remained, specifically on the points:

2. Strange references remain - reference 9 (in the previous version - 10) does not refer to Saudi Arabia, this is a letter to the editor, it refers to an article about India. I do not understand why the authors stubbornly refer to this source.

4. I did not see the flowchart in the new version of the manuscript, although the authors promised to add it.

5. This information is still missing from the new version of the manuscript.

6. The authors have added some information, but questions remain. What method of checking the data for normality was used? Which method of multiple logistic regression was used? In Table 1, the authors presented p values, but it is not clear from the table which indicators they refer to, it is necessary to change the format of the table.

7. In response to this question, the authors write that thrombolytic therapy was carried out in a hospital. How to understand then this difference - TLT in 188 patients, and hospitalization - in 156?

9. The authors misinterpret the results of the statistical analysis for actor Xa inhibitor (Rivaroxiban) and for combination therapy, they should consult a statistician.

Author Response

Dear Reviewer,

Thank you very much for your valuable comments on our article. Your insights have helped us to improve the quality of our work and we really appreciate your time and effort in providing us with your feedback.

We are pleased to inform you that we have taken into consideration all your comments and suggestions.

2. we have updated the references

4. we have added the flow chart, sorry for being missing in the previous version

5. all patient was diagnosed as PE, the meaning of provoked and unprovoked VTE was added at the end of the introduction

6. the method for checking the normality of the data was checked using the Kolmogorov-Smirnov test., and multiple logistic regression was used, the p-value results were explained in the table comment.

7.we revised the data and modify the number of hospitalized patients, TLT can’t be administered in outpatient as it requires hospitalization

9. we had revised and modified the comment on the table

Once again, we appreciate your comments and would like to assure you that we are open to any further suggestions or criticisms. We promise to take it all into account to ensure the highest quality of our work.

Sincerely,

Mostafa shaban
